# Peripheral Nerve Block for Pain Management after Total Hip Arthroplasty: A Retrospective Study with Propensity Score Matching

**DOI:** 10.3390/jcm11185456

**Published:** 2022-09-16

**Authors:** Heon Jung Park, Kwan Kyu Park, Jun Young Park, Bora Lee, Yong Seon Choi, Hyuck Min Kwon

**Affiliations:** 1Department of Orthopedic Surgery, Severance Hospital, Yonsei University College of Medicine, Seoul 03722, Korea; 2Department of Orthopedic Surgery, Yongin Severance Hospital, Yonsei University College of Medicine, Yongin 16995, Gyeonggi-do, Korea; 3Department of Anesthesiology and Pain Medicine, Severance Hospital, Yonsei University College of Medicine, Seoul 03722, Korea

**Keywords:** analgesia, pain management, peripheral nerve block, postoperative pain, total hip arthroplasty, visual analogue scale

## Abstract

This study aimed to evaluate the effect of a peripheral nerve block (PNB) on immediate postoperative analgesia and the early functional outcomes for patients who underwent total hip arthroplasty (THA). From January 2016 to August 2021, 353 patients who underwent THA were divided into two groups: the patient-controlled analgesia (PCA) group (*n* = 217) who received only intravenous (IV) analgesia, and others who received IV PCA and PNB (PCA + PNB group) (*n* = 136). After propensity score matching for age and sex, 136 patients from each group were included in the study. Primary outcomes were the visual analogue scale (VAS) at rest, activity status at postoperative 6, 24, 48 h. Secondary outcomes were functional scores by the Western Ontario and McMaster Universities Osteoarthritis (WOMAC) index, Harris Hip Score (HHS) and rescue medications used. The postoperative VAS at 6, 24, 48 h at rest and 6 h at activity were significantly lower in the PCA + PNB group (*p* = 0.000, 0.001, 0.000, 0.004 in order). There was no significant difference for postoperative 3-month HHS (*p* = 0.218), except for 3-month WOMAC index (*p* = 0.001). There were no significant differences for VAS between the PNB methods except femoral nerve block (FNB) and fascia iliaca compartment block (FICB) at postoperative activity 48 h (*p* = 0.028). There was no significant difference in the total count and amount of rescue medication (*p* = 0.091, 0.069) and difference in the quadriceps weakness was not noted. Therefore, PNB is beneficial for patients who undergo THA as it provides sufficient postoperative analgesia, especially during immediate postoperative resting pain without quadriceps weakness.

## 1. Introduction

Total hip arthroplasty (THA) is the primary, most common orthopedic surgery to alleviate hip pain due to degenerative diseases by replacing pathogenic hip joint to artificial joint, resulting in improved quality of life [1,2]. As societies age, more people are diagnosed with orthopedic degenerative diseases, and it is estimated that >400,000 THAs are annually performed in the United States [3]. However, most patients experience severe pain after THA [3,4,5]. This results in longer hospital stays, delayed ambulation and physiotherapy, low satisfaction, and a higher risk of thrombotic events [4,5]. Increased opioid consumption for pain relief can lead to gastrointestinal problems, impaired cognitive function such as delirium, urinary problems, and respiratory arrest [5,6,7,8]. The concept of enhanced recovery after surgery (ERAS) was first described by Henrik Kehlet in 1997 [9]. It is highly utilized in orthopedics, where the surgical goal is functional recovery [10,11].

A peripheral nerve block (PNB) for patients who undergo total joint arthroplasty is a recommended intervention with ERAS protocols [10,12]. PNBs provide sufficient analgesia with fewer adverse effects, such as neurological complications, nausea, and hypotension than epidural or intravenous patient-controlled analgesia (PCA) with opioids [4,5,13]. For total knee arthroplasty (TKA), femoral nerve block (FNB) is the most commonly used PNB method, and adductor canal block is an alternative choice that selectively blocks the sensory branch of femoral nerve [14,15]. Various studies have already proved the effectiveness of PNB for TKAs, making PNB an important factor in the ERAS protocol.

Compared to studies about TKAs, there are fewer studies about the effect of PNBs for THAs. There are more PNB methods used for THA compared to TKA, and they are technically more demanding; FNB, quadratus lumborum block (QLB), fascia iliaca compartment block (FICB), lateral femoral cutaneous nerve block (LFCNB), and pericapsular nerve group block (PENG) [16,17,18,19,20]. Despite known PNB methods, there are concerns about the effectiveness of a PNB because hip joint innervation is complex and motor-nerve blocks can cause quadriceps weakness. Based on previous studies, PNB such as FNB may reduce postoperative pain after THA, but most studies have focused on the analgesic effect and side effects of PNB. Studies regarding the impact of PNB for functional outcome after THA seems insufficient and none of the previous studies have addressed the pain management of PNB when adding with IV PCA.

This study was hypothesized that adding PNB to conventional IV PCA will significantly alleviate pain after THA and improve early functional outcomes. This study aimed to compare the results of PNB administered in addition to intravenous (IV) PCA and IV PCA alone for the effectiveness of postoperative pain management, the difference in functional outcomes, and the effectiveness of various PNBs after THA.

## 2. Materials and Methods

### 2.1. Data Collection

This is a retrospective study with propensity score matching to analyze the data. We reviewed the electrical medical records at a single tertiary hospital from January 2016 to August 2021 for patients aged > 20 years who underwent THA after a diagnosis of osteonecrosis of the femoral head or osteoarthritis of the hip joint (degenerative, secondary). We obtained informed consent from all the participants for accessing their data before surgery and received approval from the Institutional Review Board of Severance Hospital (4-2022-0179, approved on 6 April 2022). We included patients in this study whose data such as demographics, type of anesthesia, and American Society of Anesthesiologists (ASA) class was able to be collected by the researchers. The exclusion criteria were patients with inflammatory hip arthritis, rheumatoid arthritis, periprosthetic joint infection, history of revision surgery, patients with special devices due to severe instability, anatomical deformity, bone defects, and adverse reactions to opioids or anesthesia agents, and uncontrolled glucose levels.

With these criteria, 353 patients were included and divided into 2 groups, with 217 patients who controlled postoperative pain using only intravenous IV PCA (PCA group) and 136 patients who controlled postoperative pain using PNB and IV PCA (PCA + PNB group). Propensity score matching was performed for age and sex among these groups, and only 136 patients from each group were included in the study (Figure 1). FNB, QLB, FICB, LFCNB, and PENG were performed by experienced anesthesiologists after the surgery with a regimen of ropivacaine. Additional intravenous PCA was available for all PCA + PNB group patients.

All THA operations were performed by a single surgeon with the patient in a lateral position. All surgeries were performed using a posterolateral hip approach and the short rotators were repaired. Thirty minutes before the end of the surgery, IV fentanyl 1 μg/kg and palonosetron 0.075 mg were administered to the patient for postoperative analgesia and antiemetic effects, respectively. All patients were administered IV PCA for 48 h postoperatively, which comprised fentanyl 7 μg/kg and palonosetron 0.075 mg (total volume including saline 100 mL), delivered as a 2 mL/h background infusion and 0.5 mL doses on patient demand with a 15-min lockout time. In the ward, all patients were administered celecoxib 200 mg orally and acetaminophen 1 g intravenously every 12 h. 

All patients started postoperative exercise following the same rehabilitation protocol. Alongside this, bedside exercises (ankle pumps, quadriceps stretching, leg raising) were performed 0–6 h after the operation. Standing and walker ambulation was permitted on postoperative day 1 following the same protocol. Additional pain control was administered as rescue medication (pethidine 25 mg, pethidine 50 mg, tramadol 50 mg) through intramuscular injections.

### 2.2. Outcome Measurements

The primary outcome was pain intensity scores measured on a visual analogue scale (VAS) (0–10, with 0 = no pain and 10 = worst possible pain). VAS at rest and activity (during 45° passive flexion of the hip) were monitored for the first 48 h after surgery at 6, 24, and 48 h.

Secondary outcomes included functional outcomes measured by the Western Ontario and McMaster Universities Osteoarthritis (WOMAC) index and Harris Hip Score (HHS) at postoperative month 3 [21,22]. The total number and amount of rescue medications were also assessed.

### 2.3. Statistical Analysis

Descriptive statistics were performed, and normality distribution analysis was assessed by the Shapiro–Wilk test. Continuous variables were analyzed using the Student’s *t*-test (normal distributions) or Mann–Whitney test (non-normal distributions). Categorical variables were compared using the chi-square test. The Kruskal–Wallis test was used to evaluate non-normal distribution in the continuous variables of multiple groups. The Bonferroni correction method for post-hoc tests was conducted if necessary. *p*-value < 0.05 was considered statistically significant in all cases. An analysis of data and propensity score matching was performed using the Statistical Package for Social Science software version 26.0 (SPSS, IBM Inc., Chicago, IL, USA).

## 3. Results

The baseline characteristics of patients were compared for age, sex, body mass index (BMI), and ASA class between the PCA and PCA + PNB groups after propensity score matching (Table 1).

### 3.1. Primary Outcome

The VAS for THA were significantly lower in the PCA + PNB group compared with those in the PCA group at postoperative 6, 24, and 48 h during rest (*p* = 0.000, 0.001, and 0.000, respectively, Figure 2). VAS in the PCA + PNB group was lower at 6 h postoperatively during activity (*p* = 0.004, Figure 3). However, there was no significant difference between the VAS at postoperative 24 and 48 h at activity (*p* > 0.05, Figure 3).

### 3.2. Secondary Outcome

There were no differences between the PCA and PCA + PNB groups in the secondary outcomes for early functional outcomes with the HHS at postoperative month 3, except for the 3-month postoperative WOMAC index (*p* = 0.009) (Table 2).

We also analyzed the total count of rescue medication prescribed during the first 2 postoperative days and converted the total dose of rescue medication within this period into a total morphine equivalent dose (mg) using converting factors [23,24,25]. As a result, the average count of opioids consumed during the first two days decreased by 0.26 when a PNB was used in addition to PCA. When a PNB was administered to patients, the use of opioids decreased by a morphine equivalent of 2.3 mg; however, these results were not statistically significant (Table 3).

### 3.3. Subgroup Analysis

We conducted a subgroup analysis to review the analgesic effect of the PNB procedures. From the five PNB methods, we excluded QLB as it had been performed on a small number of patients (Table 1). We combined the FNB subgroup and the FNB + LFCNB subgroup to form one FNB group (Table 1). As a result, there were no significant differences in VAS between the PNB procedures postoperatively during rest and at postoperative 6, 24 h during activity (Table 4). When any two PNB procedures were compared at 48 h postoperatively, the VAS of the FNB group was significantly lower than the FICB group at activity (*p* = 0.028, Table 5).

## 4. Discussion

This single-center, large-sample retrospective study investigated the analgesic effect and early postoperative functional outcomes of PNB by measuring the improvement in symptoms after the addition of PNB to conventional IV PCA. The study also investigated the effect of different types of PNB on pain control. The results showed that immediate postoperative pain in the first 48 h of surgery was significantly resolved by PCB with PNB compared with IV PCA alone. The periods studied were 0–6 h, 6–24 h, and 24–48 h, and all three subgroups at rest showed significant pain control with the PNB compared to the use of IV PCA alone. The period 0–6 h at activity also showed significant pain control with PCA + PNB compared to IV PCA alone. However, there was no difference in the analgesic effect of adding PNB during activity at 24 and 48 h postoperatively. 

Alleviated resting and activity pain in the immediate postoperative period after THA surgery is the most important finding of this study. As patients experience less pain after surgery, they require fewer pain medications such as opioids, which can lower adverse effects such as gastrointestinal discomfort. Moreover, this results in decreased level of delirium, which can lead to lower accidental fall rates while admitted in the hospital and unnecessary medical expenses. The results showed that the addition of PNB can decrease the total amount of opioid consumption as rescue pain medication (Table 3).

Patients who undergo THA experience significant postoperative pain [26]. Providing sufficient pain management is crucial for early rehabilitation and physical therapy, which is important for achieving an adequate range of motion of the joint and discharge from the hospital [26]. As the number of patients undergoing THA has increased, pain management after THA has gained importance. Historically, general anesthesia was the gold standard for patients undergoing hip surgery [27]. Recent studies about anesthesia and advanced technology, such as ultrasonography and catheters, allow the anesthetist to administer spinal or epidural anesthesia as perioperative anesthesia and pain management [26,27,28]. The continuous PNB was introduced to administer postoperative analgesia to THA [27]. Numerous studies have shown various regional block methods of peripheral nerve blocks for total joint arthroplasty; however, they focused on anesthetic procedures, devices, and technical issues [27]. Recent studies have been conducted on postoperative pain control and functional outcomes of peripheral nerve blocks compared with conventional methods [26,28]. Hebl et al. (2008) showed that the posterior lumbar plexus (psoas compartment) block was superior to IV PCA alone in terms of reduced intravenous opioid consumption, opioid adverse effects (e.g., nausea, vomiting), length of hospital stay, and improved range of motion [26]. Our study showed similar results for pain control, even by adding PNB compared to IV PCA alone after propensity matching scores for age and sex. Furthermore, our study investigated the differences in the analgesic effects during postoperative periods of rest and activity. Pain scale at postoperative 0–48 h at rest, 0–6 h during activity were significantly decreased by adding PNB.

PNB provides site-specific analgesia and has fewer adverse effects than epidural or intravenous anesthesia. Various studies have reported that patients administered PNB experience less urinary retention, hypotension, and nausea or vomiting than patients administered epidural anesthesia while maintaining a similar analgesic effect [29]. All patients in this study who were administered PNB did not have major complications, e.g., quadriceps weakness.

Few studies have compared the early follow-up functional outcomes after THA. As our institute routinely measures the WOMAC index and HHS pre-operatively (one day before surgery), on postoperative day 90, and 6 months, and 1, 2, and 3 years after patients undergo THA, we were able to examine each patient’s functional outcomes after THA using these two scoring systems. We found that there were no significant differences in HHS between PCA only and PCA + PNB groups. Contrary to expectations, the WOMAC index of the PCA group at postoperative month 3 were lower than that of the PCA + PNB group. Hebl et al. (2008) reported that patients with PNB could sit up and walk significantly earlier than those who used PCA alone [26]. The reason the WOMAC index of the PCA group was better at postoperative month 3 needs further investigation. However, we believe the effect of PNB on the long-term outcome is minimal, as the duration of the anesthetic effect by the agent used in the PNB is ≤24 h. Therefore, further studies could focus on long-acting local anesthetic agents for PNB.

The Department of Anesthesiology of our institution performed five different methods of PNB in the present study as mentioned above. Rashiq et al. (2013) conducted a systematic review of studies about randomized control trials between 1990 and 2010 to study the effects of PNB on patients undergoing hip surgery. This review concluded that a combination of an obturator and lateral femoral cutaneous block had the best analgesic effect on immediate postoperative pain, and a fascia iliaca block resulted in the lowest delirium rate [30]. Wang et al. (2013) conducted a meta-analysis of the effects of pain control using an FNB and a fascia iliaca block for patients undergoing TKA or THA and reported no significant difference in pain control and the consumption of opioids between performing an FNB and a fascia iliaca block [18]. Lin et al. (2021) performed a single-center double-blinded randomized trial comparing the short-term analgesic effects of PENG and FNB. They reported that PENG had better pain control in the recovery room with better preserved quadriceps strength [31]. Other studies have shown conflicting results on the superiority of the different PNB methods [13,16,17,18,19,20,25,29]. In our study, the level of pain management did not show significant differences among various types of PNB, except FNB and FICB group at postoperative 48 h during activity.

The strength of our study is in its larger patient numbers and control by propensity score matching. We investigated the effect of analgesia based on subjective pain and the functional outcomes for patients who underwent THA. Using statistics, we investigated the superiority of pain relief from the various PNB methods available. Lastly, this study addressed the analgesic effect of the combination of PNB and IV PCA, which most previous studies have not dealt with.

This study has several limitations. First, this was a single-center, retrospective comparative study. We analyzed the pain management and outcomes of a large number of patients and performed propensity score matching to complement the biases. Since this study was not a randomized controlled study, there may be a patient’s selection bias. Second, several patients did not complete the WOMAC index and HHS during the surveys. Third, the bias from the pain scale and survey cannot be overlooked, as it is a subjective index. Fourth, the differences in the diagnosis of hip pathology were not included as a parameter. Fifth, the different types of THA implants by manufacturers were not considered.

## 5. Conclusions

PNB has remarkable benefits for immediate postoperative pain control after primary THA. The analgesic effect of PNB with IV PCA was better than conventional IV PCA alone. There was no significant difference in the functional outcomes (WOMAC index, HHS) of patients who had IV PCA and IV PCA + PNB. The analgesic effect among different PNB methods showed no significant difference. Further investigations are needed concerning the ideal, detailed regimen of PNB and various adverse effects such as in-hospital falls.

## Figures and Tables

**Figure 1 jcm-11-05456-f001:**
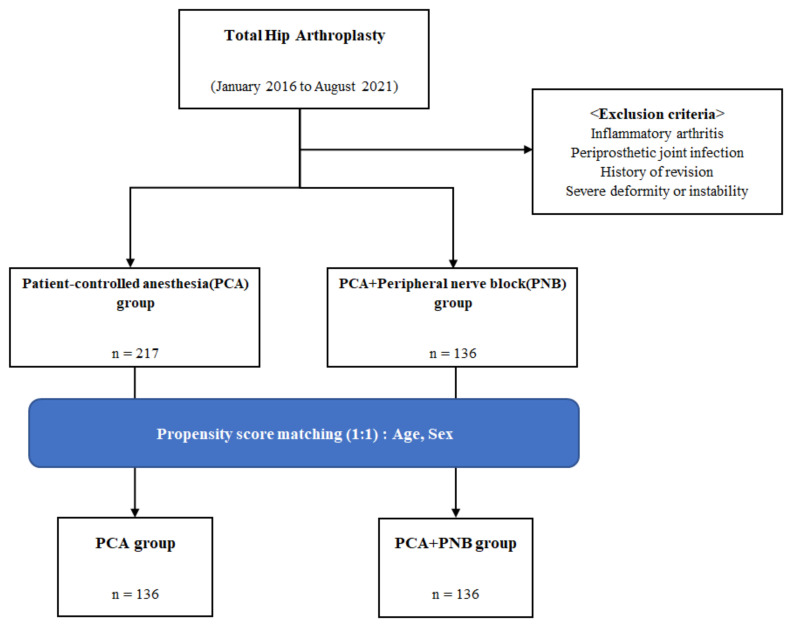
Flow chart for patient selection.

**Figure 2 jcm-11-05456-f002:**
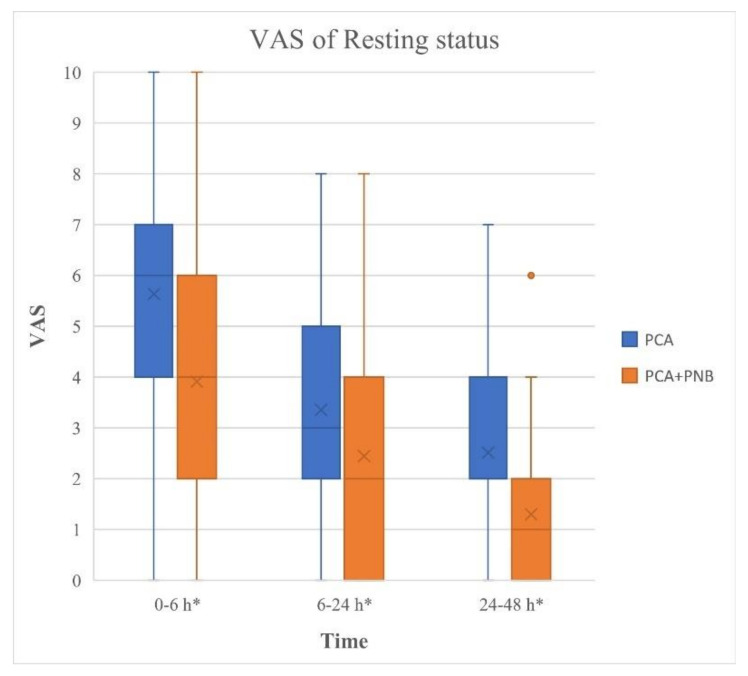
VAS during rest for patients from the PCA and PCA + PNB groups who underwent THA. * Statistically significant difference between two groups (*p* < 0.05). PCA: patient-controlled analgesia; PNB: peripheral nerve block; THA: total hip arthroplasty; VAS: visual analogue scale.

**Figure 3 jcm-11-05456-f003:**
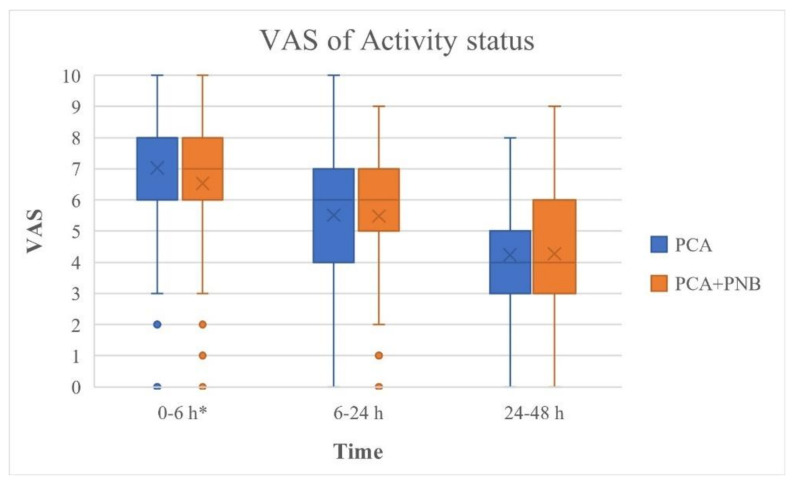
VAS during activity for patients from the PCA and PCA + PNB groups who underwent THA. * Statistically significant difference between two groups (*p* < 0.05). PCA: patient-controlled analgesia; PNB: peripheral nerve block; THA: total hip arthroplasty; VAS: visual analogue scale.

**Table 1 jcm-11-05456-t001:** Baseline characteristics of the patients undergoing total hip arthroplasty.

	Total Hip Arthroplasty		
	PCA Group (*n* = 136)	PCA + PNB Group (*n* = 136)	*p*-Value
**Age (years)**	51.7 ± 11.4	59.1 ± 13.0	0.000 *
**Sex**			
Male No. (%)	49 (36.0)	59 (43.4)	
Female No. (%)	87 (64.0)	77 (56.6)	
**BMI**	24.2 ± 3.4	25.0 ± 3.8	0.043 *
**ASA (%)**			0.199
1	21 (15.4)	8 (6.0)	
2	64 (47.1)	73 (53.7)	
3	49 (36.0)	54 (39.6)	
4	2 (1.5)	1 (0.7)	
**PNB method**			
FNB		66	
FICB		36	
PENG		29	
Others		5	
**PCA**			
IV PCA	136	NA	
IV PCA + PNB	NA	136	

Data are shown as mean ± standard for normally distributed variables. * Statistically significant difference between two groups (*p* < 0.05). PCA: patient-controlled analgesia; PNB: peripheral nerve block; BMI: body mass index; ASA: American Society of Anesthesiologists.

**Table 2 jcm-11-05456-t002:** Comparison of the functional outcomes with the WOMAC index and HHS between the PCA and PCA + PNB groups with total hip arthroplasty.

	Total Hip Arthroplasty		
	PCA Group (*n* = 136)	PCA + PNB Group (*n* = 136)	*p*-Value
**WOMAC**			
Pre-operative	50.2 ± 21.1	51.7 ± 20.1	0.562
Post-operative 3 months	14.0 ± 12.7	25.0 ± 21.3	0.009 *
**HHS**			
Pre-operative	50.7 ± 19.5	48.1 ± 20.2	0.278
Post-operative 3 months	71.7 ± 34.1	76.4 ± 18.0	0.210

Data are shown as mean ± standard for normally distributed variables. * Statistically significant difference between two groups (*p* < 0.05). PCA: patient-controlled analgesia; PNB: peripheral nerve block; WOMAC: Western Ontario and McMaster Universities Osteoarthritis index; HHS: Harris Hip Score.

**Table 3 jcm-11-05456-t003:** Comparison of rescue pain medication between the PCA and PCA + PNB groups after total hip arthroplasty.

	Rescue Medication		
	PCA Group (*n* = 136)	PCA + PNB Group (*n* = 136)	*p*-Value
**Total count**	2.05 ± 2.3	1.79 ± 2.4	0.091
**Morphine equivalent (mg)**	23.4	21.1	0.069

PCA: patient-controlled analgesia; PNB: peripheral nerve block.

**Table 4 jcm-11-05456-t004:** Subgroup analysis of VAS between the PNB groups for patients who underwent total hip arthroplasty.

	FNB (*n* = 66)	FICB (*n* = 36)	PENG (*n* = 29)	*p*-Value
0–6 h Rest	5 (1, 6)	5 (2, 6)	3 (2, 5)	0.818
0–6 h Activity	7 (5.5, 8)	7 (6, 8)	7 (6, 8)	0.684
6–24 h Rest	3 (1, 4)	3 (0, 5)	2 (0, 3)	0.161
6–24 h Activity	5 (5, 6.5)	6 (5.5, 7)	5 (4, 6)	0.052
24–48 h Rest	2 (0, 2)	0.5 (0, 2)	0.5 (0, 1.3)	0.179
24–48 h Activity	4 (2, 5)	5 (4, 6)	5 (3, 5)	0.014 *

Data are shown as median and interquartile ranges for variables that were not normally distributed. * Significant difference between two groups (*p* < 0.05). VAS: visual analogue scale; PNB: peripheral nerve block; FNB: femoral nerve block; FICB: fascia iliaca compartment block; PENG: pericapsular nerve group block.

**Table 5 jcm-11-05456-t005:** Subgroup analysis of VAS among the PNB groups, 24–48 h postoperatively during activity.

PNB Subgroup	*p*-Value	*p*-Value (Adjusted)
FNB-PENG	0.338	1.000
FNB-FICB	0.004 *	0.021 *
PENG-FICB	0.129	0.775

* Significant difference between two groups. VAS: visual analogue scale; PNB: peripheral nerve block; FNB: femoral nerve block; FICB: fascia iliaca compartment block; PENG: pericapsular nerve group block.

## Data Availability

The data used, analyzed during this study are available from the corresponding author on reasonable request.

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
