# Peer review of "Peripheral Nerve Block for Pain Management after Total Hip Arthroplasty: A Retrospective Study with Propensity Score Matching"

_jcm, 2022, doi:10.3390/jcm11185456_

Round 1

Reviewer 1 Report

 It was a pleasure to read your research on postoperative pain management after THA. This procedure is something that we always recommend with the aid from our anesthesia colleagues. The procedure clearly has its advantages over standard antalgics and this is also highlighted in your research, providing the literature with more information and data. However, I have few suggestions that require your review:

- please replace keyword 1 with an actually keyword;

- authors should provide an institutional board review number for this article? (Row 294 has phrasing errors too)

- in Figure 1 - please adjust the spaces of exclusion criteria tab. It seems you have a row empty.

- calendar dates mentioned in materials and methods are not the same with dates from Figure 1.

- approximately 40% of references are 10 years old or more; i suggest updating your literature.

Author Response

 It was a pleasure to read your research on postoperative pain management after THA. This procedure is something that we always recommend with the aid from our anesthesia colleagues. The procedure clearly has its advantages over standard antalgics and this is also highlighted in your research, providing the literature with more information and data. However, I have few suggestions that require your review:

- please replace keyword 1 with an actually keyword;
: We revised Keyword using actually keywords.

- authors should provide an institutional board review number for this article? (Row 294 has phrasing errors too)
: We added IRB approval number of our institution (4-2022-0179)

- in Figure 1 - please adjust the spaces of exclusion criteria tab. It seems you have a row empty.
: We revised Figure 1.

- calendar dates mentioned in materials and methods are not the same with dates from Figure 1.
We revised calendar dates.

- approximately 40% of references are 10 years old or more; i suggest updating your literature.
: We updated references.

Reviewer 2 Report

1.      Every author's email is written in black, without any underlining, and according to MDPI format.

2.      The abstract requires the addition of quantitative results.

3.      As the conclusion of your abstract, please provide a "take-home" message.

4.      Rearrange the keywords so that they are in alphabetical order.

5.      What makes the author's novelty in the present work? My analysis suggests that other similar previous studies properly explain the points you have brought up in the current paper related to pain management after total hip prosthesis. Please be sure to emphasize anything truly novel in this work in the introductory section.

6.      In order to highlight the gaps in the literature that the most recent research aims to fill, it is crucial to review the benefits, novelty, and limitations of earlier studies in the introduction.

7.      Line 66, do not use “We”, make it into passive.

8.      In line 40-41, the extended explanation and additional reference related to total hip arthroplasty is needed for explaining its objective for replace problematic human hip joint. The suggested reverence should be taken to substantiate this explanation as follows: Ammarullah, M. I.; Santoso, G.; Sugiharto, S.; Supriyono, T.; Kurdi, O.; Tauviqirrahman, M.; Winarni, T. I.; Jamari, J. Tresca Stress Study of CoCrMo-on-CoCrMo Bearings Based on Body Mass Index Using 2D Computational Model. Jurnal Tribologi 2022, 33, 31–8. https://jurnaltribologi.mytribos.org/v33/JT-33-31-38.pdf

9.      To improve the reader's understanding of the materials and methods section simpler for them to grasp rather than only relying on the predominate text as it currently exists, the authors could incorporate figures that illustrate the workflow of the current study.

10.   What is the basis for data selection? Is there any protocol, standard, or basis that has been followed? It is unclear since the patient is very heterogeneous with a small number. The resonance involved impacts the present result makes this study flaws. One major reason for rejecting this paper.

11.   Before moving on to the conclusion section, the present study's limitation must be added at end of the discussion section.

12.   Please discuss the further research in the conclusion section.

13.   The reference should be given additional literature from the recent five years for enriching it. MDPI literature is highly recommended.

14.   The authors occasionally created paragraphs in the entire document that were just one or two phrases long, which made the explanation difficult to understand. To make their explanation into a longer, more thorough paragraph, the authors should expand it. It is advised to use at least three sentences in a paragraph, with one serving as the primary sentence and the others as supporting phrases. See line 241-245.

15.   Due to grammatical and linguistic style issues, the authors should proofread the manuscript. For this issue, the authors would utilize the MDPI English editing service.

16.   Kindly double-check that the authors followed the MDPI format correctly, then modify the current form and recheck for any additional issues that have been discovered.

Author Response

 1.      Every author's email is written in black, without any underlining, and according to MDPI format.
: We revised e-mail address.

2.      The abstract requires the addition of quantitative results.
: We added quantitative results in Abstract.

3.      As the conclusion of your abstract, please provide a "take-home" message.
: We added “take-home” message in conclusion of abstract. (Line 29-31)

4.      Rearrange the keywords so that they are in alphabetical order.
: We revised keywords.

5.      What makes the author's novelty in the present work? My analysis suggests that other similar previous studies properly explain the points you have brought up in the current paper related to pain management after total hip prosthesis. Please be sure to emphasize anything truly novel in this work in the introductory section.
: Unlike previous studies, our study analyzed not only the pain reduction effect of peripheral nerve block but also the effect on functional outcome. Also, the comparison of pain management in THA with simultaneous IV PCA and PNB and PCA alone is different from previous studies. Therefore, we think that our study has novelty, and we added it in ‘Introduction’ section. (Line 67-71)

6.      In order to highlight the gaps in the literature that the most recent research aims to fill, it is crucial to review the benefits, novelty, and limitations of earlier studies in the introduction.
: We revised manuscript adding the limitations of earlier studies and novelty of this study. (Line 67-71)

7.      Line 66, do not use “We”, make it into passive.
: We revised manuscript. (Line 72-76)

8.      In line 40-41, the extended explanation and additional reference related to total hip arthroplasty is needed for explaining its objective for replace problematic human hip joint. The suggested reverence should be taken to substantiate this explanation as follows: Ammarullah, M. I.; Santoso, G.; Sugiharto, S.; Supriyono, T.; Kurdi, O.; Tauviqirrahman, M.; Winarni, T. I.; Jamari, J. Tresca Stress Study of CoCrMo-on-CoCrMo Bearings Based on Body Mass Index Using 2D Computational Model. Jurnal Tribologi 2022, 33, 31–8. https://jurnaltribologi.mytribos.org/v33/JT-33-31-38.pdf
: We revised manuscript using suggested reference. (Line 42-44)

9.      To improve the reader's understanding of the materials and methods section simpler for them to grasp rather than only relying on the predominate text as it currently exists, the authors could incorporate figures that illustrate the workflow of the current study.
: To make it easier to understand the outline of the study, we created a flowchart of patient selection using patient enrollment and propensity score matching statistical methods and shown in figure 1.

10.   What is the basis for data selection? Is there any protocol, standard, or basis that has been followed? It is unclear since the patient is very heterogeneous with a small number. The resonance involved impacts the present result makes this study flaws. One major reason for rejecting this paper.
: Since 2018, we have been implementing peripheral nerve block if the patient’s consent before surgery. As you said, a bias may exist because the patient is heterogeneous with a small number and the criteria are unclear, and the study is not randomized controlled study. We performed propensity score matching analysis to minimize this bias, and since this is a limitation of this study, it was added to the limitation. (Line 284-285)

11.   Before moving on to the conclusion section, the present study's limitation must be added at end of the discussion section.
: We added limitations before conclusion. (Line 282-289)

12.   Please discuss the further research in the conclusion section.
: We added further research in the conclusion section (Line 296-297)

13.   The reference should be given additional literature from the recent five years for enriching it. MDPI literature is highly recommended.
: We added more recent references.

14.   The authors occasionally created paragraphs in the entire document that were just one or two phrases long, which made the explanation difficult to understand. To make their explanation into a longer, more thorough paragraph, the authors should expand it. It is advised to use at least three sentences in a paragraph, with one serving as the primary sentence and the others as supporting phrases. See line 241-245.
: We revised manuscript as your comments. 

15.   Due to grammatical and linguistic style issues, the authors should proofread the manuscript. For this issue, the authors would utilize the MDPI English editing service.
: We performed additional English proofreading

16.   Kindly double-check that the authors followed the MDPI format correctly, then modify the current form and recheck for any additional issues that have been discovered.
: Thank you for your comment. We checked it.

Round 2

Reviewer 2 Report

My recommendation is to accept in the present form.